# An Antioxidant Supplement Function Exploration: Rescue of Intestinal Structure Injury by Mannan Oligosaccharides after *Aeromonas hydrophila* Infection in Grass Carp (*Ctenopharyngodon idella*)

**DOI:** 10.3390/antiox11050806

**Published:** 2022-04-20

**Authors:** Zhi-Yuan Lu, Lin Feng, Wei-Dan Jiang, Pei Wu, Yang Liu, Xiao-Wan Jin, Hong-Mei Ren, Sheng-Yao Kuang, Shu-Wei Li, Ling Tang, Lu Zhang, Hai-Feng Mi, Xiao-Qiu Zhou

**Affiliations:** 1Animal Nutrition Institute, Sichuan Agricultural University, Chengdu 611130, China; luzhiyuan@stu.sicau.edu.cn (Z.-Y.L.); fenglin@sicau.edu.cn (L.F.); wdjiang@sicau.edu.cn (W.-D.J.); wupei0911@sicau.edu.cn (P.W.); 11081@sicau.edu.cn (Y.L.); xwjin@sicau.edu.cn (X.-W.J.); hmren@sicau.edu.cn (H.-M.R.); 2Fish Nutrition and Safety Production University Key Laboratory of Sichuan Province, Sichuan Agricultural University, Chengdu 611130, China; 3Key Laboratory for Animal Disease-Resistance Nutrition of China Ministry of Education, Sichuan Agricultural University, Chengdu 611130, China; 4Animal Nutrition Institute, Sichuan Academy of Animal Science, Sichuan Animtech Feed Co., Ltd., Chengdu 610066, China; ksy_cd@163.com (S.-Y.K.); lishuwei84511614@126.com (S.-W.L.); tingling@vip.163.com (L.T.); 5Healthy Aquaculture Key Laboratory of Sichuan Province, Tongwei Co., Ltd., Chengdu 610041, China; luzhangtw@163.com (L.Z.); mihaifengtw@163.com (H.-F.M.)

**Keywords:** permeability, intestine, tight junction, adherent junction, antioxidant capacity

## Abstract

Mannan oligosaccharides (MOS) are a type of functional oligosaccharide which have received increased attention because of their beneficial effects on fish intestinal health. However, intestinal structural integrity is a necessary prerequisite for intestinal health. This study focused on exploring the protective effects of dietary MOS supplementation on the grass carp’s (*Ctenopharyngodon idella*) intestinal structural integrity (including tight junction (TJ) and adherent junction (AJ)) and its related signalling molecule mechanism. A total of 540 grass carp (215.85 ± 0.30 g) were fed six diets containing graded levels of dietary MOS supplementation (0, 200, 400, 600, 800 and 1000 mg/kg) for 60 days. Subsequently, a challenge test was conducted by injection of *Aeromonas hydrophila* for 14 days. We used ELISA, spectrophotometry, transmission electron microscope, immunohistochemistry, *q*RT-PCR and Western blotting to determine the effect of dietary MOS supplementation on intestinal structural integrity and antioxidant capacity. The results revealed that dietary MOS supplementation protected the microvillus of the intestine; reduced serum diamine oxidase and d-lactate levels (*p* < 0.05); enhanced intestinal total antioxidant capacity (*p* < 0.01); up-regulated most intestinal TJ and AJ mRNA levels; and decreased GTP-RhoA protein levels (*p* < 0.01). In addition, we also found several interesting results suggesting that MOS supplementation has no effects on *ZO-2* and *Claudin-15b*. Overall, these findings suggested that dietary MOS supplementation could protect intestinal ultrastructure, reduce intestinal mucosal permeability and maintain intestinal structural integrity via inhibiting MLCK and RhoA/ROCK signalling pathways.

## 1. Introduction

According to the Food and Agriculture Organization/World Health Organization (FAO/WHO) definition, probiotics are ‘live microorganisms that, when administered in adequate amounts, confer a health benefit on the host’ [1]. Many studies indicate that probiotics (such as *Bacillus* sp., *Lactobacillus* sp., and *Enterococcus* sp.) contribute to improving fish gut health and structural integrity [2,3,4,5,6]. Based on the available literature, probiotics play a beneficial role by optimizing the intestinal flora and improving the apparent digestibility of feed and fermentation of water-soluble fiber (such as oligosaccharides) in food, and so on [3,7,8,9]. In addition, the beneficial effects of short-chain fatty acids (acetic, propionic and butyric), probiotic metabolites, on intestinal structure and function have been reported [10,11]. Prebiotics are like nutrients for several beneficial bacteria and are usually not affected by digestive enzymes in the gut, but they can be used by probiotics to promote their growth [12]. Therefore, the application of prebiotics in aquaculture is considered an effective solution to improve feed conversion efficiency and promote fish growth [13].

In recent decades, aquaculture is the animal production industry that has been showing the highest rate of growth around the world, and it benefits from the modern intensive aquaculture technology [14,15]. Nevertheless, the development of intensive aquaculture activities has been frequently accompanied by disease outbreaks, which has brought huge economic losses to the fish-farming industry [16]. Recently, prebiotic additives have attracted much attention because of their beneficial effects on aquaculture environments and aquatic animal health [17]. Although they are a common prebiotic, mannan oligosaccharides (MOS) are recommended as aquaculture additives because of their excellent antioxidant properties [18]. The intestine is a functional organ in fish, and is essential for growth and development [19,20]. Furthermore, the structural integrity of the intestine is very important because it provides an essential barrier, resisting the infection of harmful substances (such as pathogens, toxins, and parasites) from the foreign environment and food [21,22]. To the best of our knowledge, there have been sporadic studies on the effects of MOS on fish intestinal structural integrity. Limited studies suggest that MOS could better preserve tight junction structure in European sea bass (*Dicentrarchus labrax*) and regulate the mRNA levels of *E-cadherin* and *Claudin-3* in Rainbow Trout (*Oncorhynchus mykiss*) [23,24]. Despite these studies, however, the specific mechanism of MOS on structural integrity of the intestine has not been systemically investigated. Therefore, a systematic and in-depth investigation into the effect of MOS on the intestinal structural integrity of fish is necessary.

The integrity of intestinal structure is established and maintained by apical junctional complexes (AJCs) between epithelial cells in zebrafish [25]. Furthermore, a study on humans showed that disturbed intestinal integrity might lead to increased intestinal permeability, which could be regulated by AJC [26]. In general, serum d-lactate and diamine oxidase (DAO) have been used as the main markers of intestinal permeability [27]. As we know, intestinal AJC is a specialized intercellular junctional complex, formed by tight junction proteins (TJ) (such as Zonula Occludens (ZO), Occludin and Claudins) and adherent junction proteins (AJ) (cadherins) [28]. Moreover, according to the data reviewed, *RhoA*/*ROCK* signalling was found to regulate the AJC [29]. Up to now, only two studies have reported that MOS can adjust the AJC (such as *Claudin-3* and *E-cadherin*) in fish [23,24]. However, except for *Claudin-3* and *E-cadherin*, no study has found the effects of MOS on other tight junction proteins and adherent junction proteins in fish intestines, or related mechanisms. According to our previous study, butyrate levels were increased by adding MOS to the intestine of grass carp [30]. Studies showed that butyrate is involved in the regulation of AJ (*Cadherin-1*) and TJ (*Occludin*, *Claudin-12*, *Claudin-15*) in Gilthead Sea bream (*Sparus aurata*) [31,32]. The evidence reviewed above seems to suggest that there might be a correlation between dietary MOS supplementation and integrity of intercellular structure associated with AJCs as well as the related signaling pathways in the intestine of fish, which warrants investigation.

Grass carp is one of the most important aquaculture species, with great economic value and worldwide distribution (FAO, 2020). Our previous series of studies not only found that MOS has excellent antioxidant capacity, but also effectively promotes fish growth and intestinal development [30,33]. The present study highlights more research on the effect of MOS on intestinal AJCs and the relationship between intestinal antioxidant capacity and AJCs of grass carp. Therefore, we systematically carried out this study to explore the effects of MOS on intestinal permeability, total antioxidant capacity, tight junction- and adherent junction-related parameters. This work describes an elegant mechanism of how intestinal structural integrity changes in fish protected by MOS can be maintained via the apical junctional complex under *Aeromonas hydrophila* (*A. hydrophila*) infection. In addition, this study determined optimal MOS supplementation based on related indicators of intestinal structural integrity, which provides a valuable reference for commercial feed preparation.

## 2. Materials and Methods

### 2.1. Diets

The basic diet formula is shown in Appendix A. In brief, we used fish meal, concentrated soy protein and gelatin to provide protein source, and fish oil and soybean oil to provide a lipid source. Six treatment diets were used in this experiment of 0, 200, 400, 600, 800, 1000 mg/kg MOS (Sciphar Hi-Tech Industry, Xi’an, China, purity: 99.12%), as previously described [30]. In short, the feed preparation method involves fully mixing different components, and then extrusion and granulation through a feed-making machine. Then, natural air drying and storing at 4 °C are performed before feeding.

### 2.2. Animals

The present study used the same growth trial as that of our previous study [30]. In brief, a total of 540 healthy grass carp were provided from Tong Wei farm (Chengdu, China). The average weight of fish was 215.85 ± 0.30 g. To adapt grass carp to the environment, one month of domestication was performed before the formal trial began. Thereafter, all fish were randomly distributed to 18 experimental cages (1.4 m L × 1.4 m W ×1.4 m H) with working volume. There were 30 fish per cage. Mesh discs (100 cm diameter) were used to assemble leftover diets per cage. The growth test lasted for 60 days (feeding 4 times/day). After feeding for 30 min, leftover diets were collected, dehydrated and weighed. The current study and methods were approved by the Animal Care Advisory Committee of Sichuan Agricultural University (permit no. LZY-2018114005).

### 2.3. Challenge Trial

After the domestication period, we conducted a 60-day growth trial using diets with different MOS supplemental levels. Detailed environment parameters are as follows: suitable dissolved oxygen concentrations (>6.0 mg/L); the water temperature was in a range of 26.5–30.5 °C; and the pH was at 7.5 ± 0.3 with natural light conditions (natural long day in summer: 14 h light/10 h dark). The 14-day challenge trial was performed by us after the 60-day growth trial. Briefly, we selected 15 fish and each treatment was conducted for infection of *A. hydrophila*, which was kindly provided by the Department of Veterinary Medicine of Sichuan Agricultural University. The infection method adopted in this experiment is intraperitoneal (IP) injection, and the concentration of the injection is 2.5 × 10^8^ colony-forming units (CFU)/mL as previously described [30]. The environmental parameters are the same those in as the growth trial. At the end of the challenge trial, the intestine samples were quickly segmented (proximal intestine (PI), middle intestine (MI) and distal intestine (DI)) and stored at −80 °C for further detection.

### 2.4. Biochemical Analysis

Blood samples were collected from the tail before dissection, followed by low-speed centrifugation at 845× *g* for 10 min to obtain serum samples for the DAO and d-lactate concentration analysis (Beijing Qisong Biotechnology Co., Ltd., Beijing, China). Intestinal samples (PI, MI and DI) were prepared into 10% tissue homogenate for the total antioxidant capacity (T-AOC) analysis (Nanjing Jiancheng Bioengineering Institute, Nanjing, China). All biochemical indicators were operated according to the instructions of the commercially accessible kit.

### 2.5. Transmission Electron Microscope (TEM)

TEM determination was performed by Lilai Biotechnology (Sichuan, China). Samples from different groups and different intestinal segments were quickly fixed in 2.5% glutaraldehyde in cacodylate buffer (Lilai Biotechnology, Sichuan, China) after collection, and then were post-fixed in 1% osmium tetroxide (Leica, Munich, Germany). Subsequently, all samples were dehydrated in graded series of acetone solution and embedded in epoxy resin (Epon812, Beijing Zhongjingkeyi Technology Co., Ltd., Beijing, China), refer to Eikelberg et al. [34], with appropriate adjustment. Then, 50 nm ultrathin slices of intestinal sample were prepared using an ultrathin slicing machine (Leica EM UC7 ultramicrotome, Leica, Munich, Germany). For contrast enhancement, uranium acetate and lead citrate were used for double staining under room temperature (Beijing Zhongjingkeyi Technology Co., Ltd., Beijing, China). Image acquisition and visualization were acheieved using a JEM-1400PLUS TEM (Jeol, Tokyo, Japan).

### 2.6. Immunohistochemistry (IHC)

Collected intestinal samples were cleaned with PBS solution and quickly fixed in 4% paraformaldehyde buffer for immunohistochemical staining. Paraffin section preparation of intestinal samples was performed by Lilai Biotechnology (Sichuan, China). In short, the fixed samples were dehydrated in graded series of ethanol, then the dehydrated samples were embedded in paraffin. The paraffin block of each specimen was cut into 4 μm-thick sections. IHC was performed by the streptavidin–biotin–peroxidase complex method (SABC), and 3,3-diaminobenzidine tetrahydrochloride (DAB) was used as a chromogen. The IHC kits (SABC kit) were purchased from BOSTER Biological Technology (SA1028, Wuhan, China). The information on antibodies used for IHC is as follows: *ZO-1* (A0659, 1:100, ABclonal) and Occludin (A2601, 1:100, ABclonal). The detailed steps followed were according to reagent instructions with appropriate adjustment. In brief, paraffin sections were dewaxed with xylene and rehydrated with a graded series of ethanol solution, then, endogenous peroxidase activity was removed using 3% hydrogen peroxide. Then, the antigen was repaired using microwaves with EDTA-repair solution and incubated with 5% BSA blocking solution. Then, primary antibody was incubated overnight. After that, sections were made for biotin labeling and SABC incubation, followed by DAB coloration and hematoxylin restaining. Finally, all sections were used in a graded series of ethanol solution for dehydration, made transparent with xylene, and neutral gum was used for sealing. The prepared IHC slices were dried overnight (60 °C), then image acquisition and visualization were performed using a light microscope (TS100, Nikon, Tokyo, Japan). *ZO-1* and Occludin expression were quantified by integrated optical density (IOD) with Image Pro Plus 6.0 (Media Cybernetics, Inc., Rockville, MD, USA).

### 2.7. Quantitative Real-Time PCR (qRT-PCR)

*q*RT-PCR was conducted using a standard procedure from our laboratory, as previously described [30]. In brief, the total RNA of intestine (three segments) samples was extracted. Subsequently, RNA quality and purity were determined and reverse transcribed into cDNA, with the β-actin and GAPDH used as a reference gene. *q*RT-PCR was performed in a 10 μL reaction solution using a QuantStudio 5 Real-Time PCR System (Thermo Fisher Scientific™, Waltham, MA, USA). The specific primers for *q*RT-PCR are listed in Appendix A. The calculation method is as previously described [35].

### 2.8. Western Blotting

RhoA GTPase activity was measured by pull-down analysis (Cytoskeleton, Denver, CO, USA), as described by Wei et al. [27]. Briefly, for prepared protein homogenate, we used a high-throughput tissue grinder (SCIENTZ-48), and then the supernatant was obtained by centrifugation (14,000× *g*, 4 °C for 3 min). Part of the supernatant was used to determine the total RhoA activity and the rest was used to determine GTP-RhoA activity. Preprocessing is necessary to determine the GTP-RhoA. In short, the rest of the supernatant was gathered and incubated in binding domain (RBD)-GST bead aliquots on a rotator for 1 h, then washed with washing buffer, and then the supernatant for GTP-RhoA was detected. The samples were then subjected to electrophoresis, membrane transfer, blocking, and primary and second antibody incubation, as described by Wei et al. [27]. Finally, protein activity was visualized through an ECL kit (Beyotime Biotechnology, Shanghai, China).

### 2.9. Statistical Analysis

Data were examined for homogeneity and normal distribution by using Levene’s test and Shapiro–Wilk test, respectively. Biochemical analysis, gene expression and protein-level data were analyzed by one-way ANOVA with Tukey’s multiple comparisons test, and immunohistochemical data (IOD) were analyzed by Student’s *t*-test. Significance was declared at *p* < 0.05. Data analyses were conducted by PROC MIXED of SAS software version 9.4 (SAS Institute Inc., Cary, NC, USA, 2004), as previously described [30]. Orthogonal polynomial contrasts were used to assay the quadratic effect of MOS supplementation. Biochemical, gene expression, IOD and correlation data visualization we conducted using *R* Studio v1.2.5033 (running *R* v4.0.2, *R* Studio Inc., Boston, MA, USA) and Hiplot platform (https://hiplot.com.com, accessed on 24 March 2022) [33]. A summary mechanic diagram was drawn by Figdraw (www.figdraw.com, accessed on 24 March 2022).

## 3. Results

### 3.1. Growth and Disease-Resistance Phenotypes

The current study used the same growth trial as our previous work on grass carp [30]. After 60 days of growth trial, fish growth performance (final body weight (FBW), percent weight gain (PWG) and specific growth rate (SGR)) showed a quadratic effect (*p* < 0.01) with MOS supplementation compared with the control group, where the FBW, SGR and PWG of the optimal group (400 mg/kg MOS) were increased by 21.59%, 16.24% and 31.34%, respectively. After being challenged with A.hydrophila, the survival rate of fish in all groups was 100%, the enteritis morbidity and red-skin morbidity showed a quadratic effect with MOS supplementation (*p* < 0.05), compared with the control group, and the enteritis morbidity and red-skin morbidity of the optimal group (400 mg/kg MOS) were decreased by 53.45% and 42.56%, respectively [30,33].

### 3.2. Ultrastructural Observation of Intestine

To obverse the effect of MOS on fish intestine and apparent symptoms after *A. hydrophila* challenge, we chose the method of IP injection of bacterial solution, and the ultrastructural observation is shown in Figure 1A,B. Through transmission electron microscopy, we found that, as expected, the control group showed villus that were visibly disorganized and structurally destroyed, and the gaps between epithelial cells were large. Compared with the control group, the intestinal structure of the MOS group was more complete, with dense villus and smaller gaps between intestinal epithelial cells.

### 3.3. Intestinal Permeability Parameters

To explore the effect of MOS on intestinal structural integrity, DAO and d-lactate in serum were investigated. Commercial kits were used for testing, and the results are shown in Figure 1C,D. Our results showed that DAO concentration exhibited a quadratic effect (*p* < 0.05) with MOS supplementation. Compared with the control group, the MOS group (600 mg/kg) showed the lowest concentration. Similarly, d-lactate concentration also showed a quadratic effect (*p* < 0.01) with MOS supplementation. Compared with the control group, the MOS group (600 mg/kg) also showed the lowest concentration.

### 3.4. Intestinal Total Antioxidant Capacity (T-AOC) and Oxidative Related Biomarkers

To study the effect of MOS on the total antioxidant capacity of the intestine, we studied T-AOC of the PI, MI and DI, respectively. As is shown in Figure 1E–G, the current study data showed that in the PI MI and DI, the response of the T-AOC to the increasing level of MOS was quadratic (*p* < 0.05, *p* < 0.01, *p* < 0.05). The MOS group (400 mg/kg) exhibited the optimal response in antioxidant capacity in the PI and DI, and 600 mg/kg MOS in the MI. In our previous study, we also found that after being challenged with *A. hydrophila*, the content of ROS and oxidative damage biomarker (malonaldehyde (MDA)) in the three intestinal segments of grass carp showed significant quadratic effects with MOS supplemental levels (*p* < 0.01) [30]. Particularly, compared with the control group, ROS and MDA contents reached the lowest levels at the 400 mg/kg supplementation level. The ROS (% DCF florescence) of groups 1–6 were 100.00 ± 8.78, 67.33 ± 5.01, 48.19 ± 3.02, 52.61 ± 4.05, 65.74 ± 3.71 and 71.63 ± 4.76, *p*_quadratic_ < 0.01; 100.00 ± 8.78, 53.17 ± 3.89, 35.30 ± 3.24, 44.77 ± 4.12, 51.84 ± 4.15 and 69.88 ± 2.62, *p*_quadratic_ < 0.01; 100.00 ± 7.41, 56.72 ± 5.02, 43.91 ± 3.78, 52.40 ± 2.65, 63.57 ± 5.95 and 71.72 ± 5.02, *p*_quadratic_ < 0.01, in the PI, MI and DI, respectively. The MDA (nmol/g tissue) of groups 1–6 were 18.63 ± 1.57, 13.15 ± 1.01, 10.84 ± 0.91, 11.88 ± 0.62, 12.69 ± 0.88 and 13.42 ± 0.98, *p*_quadratic_ < 0.01; 14.88 ± 0.70, 14.47 ± 1.03, 11.67 ± 0.23, 12.23 ± 0.73, 12.52 ± 0.56 and 15.05 ± 0.47, *p*_quadratic_ < 0.01; and 18.38 ± 0.55, 14.44 ± 0.95, 12.23 ± 0.93, 13.91 ± 1.09, 14.77 ± 0.85 and 16.09 ± 0.78, *p*_quadratic_ < 0.01, in PI, MI and DI, respectively.

### 3.5. Immunohistochemical Analysis of Intestinal Tight Junction Protein

To observe the effect of MOS on intestinal epithelium after bacterial challenge, we performed immunohistochemical analysis of key tight junction proteins (*ZO-1* and Occludin) in 0 mg/kg, 400 mg/kg and 1000 mg/kg groups. We quantified the IOD of Occludin and *ZO-1* through Image Pro Plus 6.0 software for analysis, and the results are shown in Figure 2 and Figure 3, separately. In Figure 2, compared with the control group, the Occludin IOD of 400 mg/kg and 1000 mg/kg groups were significantly increased in three intestinal segments (*p* < 0.05). In Figure 3, compared with the control group, the *ZO-1* IOD of the 400 mg/kg group was significantly increased in three intestinal segments (*p* < 0.05), whereas the 1000 mg/kg group showed a significant increase only in DI. Additionally, the *ZO-1* IOD of the 1000 mg/kg group showed no significant difference compared with the control group in the PI and MI (*p* > 0.05).

### 3.6. Intestinal AJCs Gene Expression

To investigate the effect of MOS on the structural integrity of different segments, we systematically measured the expression of AJCs-related genes’ expression. As we know, the tight junction (TJ) and adherens junction (AJ) are the main components of AJCs [36]. In Figure 4, the response of the TJs and AJs (except *ZO-2* and *Claudin-15b*) in the PI, MI and DI, to the increasing level of MOS showed a quadratic effect (*p* < 0.05). In the PI, compared with the control, the gene-expression fold changes were as follows: *ZO-1* (0.89), *Occludin* (0.72), *JAM-A* (1.16), *Claudin-b* (0.96), *Claudin-c* (0.37), *Claudin-f* (0.70), *Claudin-3c* (0.75), *Claudin-7a* (0.96), *Claudin-7b* (0.51), *Claudin-11* (0.79), *Claudin-12* (0.88), *Claudin-15a* (0.96), *E-cadherin* (0.77), *β-catenin* (0.49), *Nectin* (0.90), and *Afadin* (0.86) were obviously up-regulated in the MOS group (400 mg/kg, *p* < 0.05), and *α-catenin* (0.77) was obviously up-regulated in the MOS group (600 mg/kg, *p* < 0.05). In the MI, compared with the control, *Claudin-7b* (0.73) was obviously up-regulated in the MOS group (200 mg/kg, *p* < 0.05), *Occludin* (0.54), *Claudin-f* (0.62), *Claudin-7a* (0.54), *Claudin-11* (1.11), *Claudin-12* (0.55), *Claudin-15a* (0.71), *E-cadherin* (1.04), *β-catenin* (1.07), *Nectin* (0.73), and *Afadin* (0.76) were obviously up-regulated in the MOS group (400 mg/kg, *p* < 0.05), and *ZO-1* (1.08), *JAM-A* (0.94), *Claudin-b* (0.47), *Claudin-c* (0.84), *Claudin-3c* (0.82), and *α-catenin* (1.03) were obviously up-regulated in the MOS group (600 mg/kg, *p* < 0.05). In the DI, compared with the control, *E-cadherin* (0.83) was up-regulated with MOS up to 200 mg/kg (*p* < 0.05), *Occludin* (0.76), *Claudin-b* (0.96), *Claudin-c* (0.89), *Claudin-f* (0.65), *Claudin-7a* (1.06), *Claudin-12* (0.81), *Claudin-15a* (0.73), *α-catenin* (1.10), and *Afadin* (0.88) were obviously up-regulated in the MOS group (400 mg/kg, *p* < 0.05), *ZO-1* (0.97), *JAM-A* (1.07), *Claudin-3c* (0.99), *Claudin-7b* (0.58), and *β-catenin* (0.74) were obviously up-regulated in the MOS group (600 mg/kg, *p* < 0.05), and *Claudin-11* (0.58) and *Nectin* (0.68) were obviously up-regulated in the MOS group (800 mg/kg, *p* < 0.05). Our data show that dietary MOS significantly increased the gene expression of most TJs and AJs in three intestinal segments.

### 3.7. Correlation Analysis

To clarify the relationship between AJCs (TJs and AJs) and involved signal molecules, we performed correlation analysis for these AJCs genes (Figure 5). There was a strong and moderate negative correlation (*R* > 0.7, strong correlation; 0.5 < *R* < 0.7, moderate correlation) between *MLCK* expression and the majority of studied TJs (except *ZO-1* and *Claudin-15b*, *R* < 0.5, weak correlation) in the PI and MI, whereas there was a more moderate negative correlation between MLCK expression and the majority of studied TJs in the DI (*R* > 0.5). Furthermore, the studied AJs showed a strong and moderate negative correlation with RhoA mRNA levels in three intestinal segments (*R* > 0.5). In addition, to explore whether the maintenance of intestinal epithelial structural integrity by MOS is related to the antioxidant capacity, we conducted correlation analysis between AJCs and intestinal antioxidant capacity (Figure 5). According to the results, the majority of studied AJCs gene expressions displayed a strong positive correlation with T-AOC in three intestinal segments (except *ZO-2* and *Claudin15b*, *R* < 0.5, weak correlation).

### 3.8. Key Signaling Molecule Protein Abundance and Related Signal Molecules’ Expression

To further study the specific mechanism of MOS-affected AJCs of fish intestines under the condition of pathogen infection, the gene expression of MLCK and RhoA and related signal molecules were measured. As is known, MLCK and RhoA are important signal molecules for regulating AJCs [16,24,37]. In Figure 5, the response of MLCK, RhoA, ROCK and NMII in the PI, MI and DI, to the increasing level of MOS displayed a quadratic effect (*p* < 0.05). In the PI, compared with the control, the gene expression fold changes were as follows: ROCK (0.29) was obviously down-regulated in the MOS group (200 mg/kg, *p* < 0.05), RhoA (0.58) was obviously down-regulated in the MOS group (400 mg/kg, *p* < 0.05), MLCK (0.58) and NMII (0.35) were obviously down-regulated in the MOS group (600 mg/kg, *p* < 0.05). In the MI, compared with the control, RhoA (0.54), ROCK (0.39) and NMII (0.66) were obviously down-regulated in the MOS group (400 mg/kg, *p* < 0.05), and MLCK was obviously down-regulated in the MOS group (600 mg/kg, *p* < 0.05). In the DI, compared with the control, MLCK (0.56), RhoA (0.57) and ROCK (0.63) were obviously down-regulated in the MOS group (400 mg/kg, *p* < 0.05) and NMII (0.66) was obviously down-regulated in the MOS group (600 mg/kg, *p* < 0.05).

To further validate signal molecules’ expression, we further examined GTP-RhoA and Total-RhoA in three intestinal segments by Western blot analysis. As shown in Figure 6, the response of the GTP-RhoA activity in the PI, MI and DI to the increasing level of MOS displays a quadratic effect (*p* < 0.05). Compared with the control, GTP-RhoA (0.38, 0.41, 0.39-fold change) was significantly down-regulated with MOS up to 600, 400 and 400 mg/kg (*p* < 0.05, *p* < 0.05 and *p* < 0.05) in the PI, MI and DI. As expected, this result for protein expression was consistent with gene expression.

## 4. Discussion

The current study used the same growth trial as our previous work on grass carp [30], which is a part of a larger study conducted to investigate the protective effect of the fish intestinal structure by MOS supplementation. Hence, we have confirmed that suitable MOS could increase fish growth and alleviate damage caused by bacterial challenge in multiple functional organs (such as head-kidney, spleen and skin). Most researchers consider fish growth and development to correlate strongly with intestinal health [17,38,39]. However, the integrity of the intestine is essential for maintaining normal intestinal health [40]. Although our previous study also confirmed that MOS can enhance intestinal antioxidant enzymes’ activity under *A. hydrophila* challenged conditions [30], this is still insufficient for understanding the protection of intestinal structural integrity by MOS. Consequently, to obtain further evidence for the protective effect of MOS on the integrity of fish intestine, we conducted a systematic study of AJCs and further examined the possible role of antioxidant capacity.

### 4.1. MOS Supplementation Protected Fish Intestinal Structural Integrity

MOS, a classical prebiotic derived from the *Saccharomyces cerevisiae* cell wall, is widely used in animal production [41]. An earlier study concluded that MOS has functional groups that can bind radicals [42]. We also demonstrated the excellent free-radical-scavenging ability of MOS in in vitro experiments [33]. Therefore, MOS is considered an excellent natural antioxidant, and is widely used in human and animal health [38,43].

#### 4.1.1. MOS Protects Intestinal Epithelial Ultrastructure

The intestinal epithelium is the first barrier of the intestine and consists mainly of epithelial cells and mucus [44]. There are special structures of intestinal microvilli in the outermost layer of intestinal epithelial cells, which can increase the surface area of food and intestinal contact and improve the efficiency of intestinal absorption of nutrients [45]. In this study, we observed by TEM that the intestinal epithelium was significantly damaged and the microvilli were severely injured after challenge with *A. hydrophila*. In addition, the gap between the epithelial cells was enlarged and the connective structure was destroyed. Compared with the control group, the integrity of intestinal villi in the MOS group (400 mg/kg) was better and the intercellular space was tighter. Our results indicate that the intestinal epithelial structure of fish can be destroyed under *A. hydrophila* challenged conditions, which is consistent with previous results on the intestinal ultra-microstructure of grass carp [46]. Therefore, we agreed that when the surface structure is damaged, intestinal digestion and absorption is greatly reduced, which further affects the growth and development of fish. Additionally, the addition of MOS to the diet can significantly alleviate this injury and protect the intestinal epithelial structure, thus ensuring the growth performance of fish. The growth data in our previous study also showed that the growth performance of the *A. hydrophila* group was significantly decreased, and that that of the MOS group was significantly higher than the *A. hydrophila* group, which supported our results [30]. Although differences in basic morphology of intestinal villi between the *A. hydrophila* group and the MOS group were observed by TEM, more representative phenotypes are needed to support this. Therefore, we further examined the effect of intestinal epithelial biomarkers to verify this.

#### 4.1.2. MOS Saves the Structural Integrity of Epithelial Cells

As mentioned above, DAO and d-lactate are important biomarkers for the structural integrity of intestinal epithelial cells. Under normal conditions, the levels of DAO and d-lactate in serum are very low. However, when the intestinal epithelial barrier is damaged, the DAO and d-lactate contents in the intestinal epithelial cells can be released and subsequently enter the blood, resulting in high levels of both of them in serum [47,48]. Previous studies in our laboratory on grass carp also confirmed that *A. hydrophila* challenge can significantly increase DAO and d-lactate content in serum [27]. The serum results in this study showed that DAO and d-lactate levels in the MOS group were significantly decreased compared with the control group, indicating that the integrity of the cell structure in the MOS group was better protected under *A. hydrophila* challenged conditions. Based on DAO and d-lactate, the optimal MOS levels were calculated as 623.9 and 644.7 mg/kg by a quadratic regression model. This supplemental level is significantly higher than the optimal supplemental level calculated based on growth performance (428.5 mg/kg MOS) [30]. These findings indicated that more MOS might be required for fish intestinal health. As far as we know, *A. hydrophila* attack toxicity usually causes the fish gut to produce excess free radicals, which could further induce oxidative damage [49,50]. However, the T-AOC of the intestine is an important phenotypic indicator that could reflect excessive free-radical-scavenging capacity [51]. Therefore, we next examined the T-AOC of different intestinal segments.

#### 4.1.3. MOS Enhances the Intestinal Total Antioxidant Capacity (T-AOC)

Fish antioxidant capacity is mainly reflected by the antioxidant system, which is composed of enzymatic antioxidants and non-enzymatic antioxidants [52]. In our previous studies, it has been found that MOS can significantly improve the activities of several antioxidant enzymes (such as SOD, CAT, GPx and GST), but cannot comprehensively reflect the total antioxidant capacity of the intestine [30]. In this study, our results showed that MOS significantly promoted the T-AOC of three intestinal segments, which was consistent with our previous results. These results indicated that MOS could effectively enhance fish intestinal antioxidant capacity. Similar results were found in other aquatic animals, such as Chinese mitten crab (*Eriocheir sinensis*) and Nile tilapia (*Oreochromis niloticus*), that dietary MOS supplementation significantly increased intestinal T-AOC and antioxidant enzyme levels (GPx and CAT) [53,54]. Otherwise, it has been reported that the basic constituents of MOS include hydroxyl groups, which could react with ROS (such as superoxide anion, hydrogen peroxide and hydroxyl radical) [55]. This has been confirmed in our previous study in vitro where MOS could increase free-radical-scavenging rate [33]. Otherwise, our in vivo experiment also demonstrated that optimal MOS supplementation (400 mg/kg) could obviously decrease ROS and MDA contents in the intestine (ROS decreased by 51.81%, 64.70% and 56.09% in the PI, MI and DI, respectively; MDA decreased by 41.81%, 21.57% and 33.46% in the PI, MI and DI, respectively) [30]. Therefore, we believe that dietary MOS can enhance fish intestine antioxidant capacity through two aspects: (1) MOS itself is an excellent natural antioxidant with special functional groups able to remove excess free radicals. (2) MOS works by activating the antioxidant system in the fish gut. However, it is worth noting that bacteria can not only produce excessive free radicals, but their main surface component, lipopolysaccharides (LPS), are also an important accomplice (Gram-negative bacteria) [56,57]. Several studies have already shown that LPS and ROS can destroy the tightly connected structure of fish intestinal epithelial cells [46,50,58]. Therefore, we performed IHC analysis on the main junction proteins Occludin and *ZO-1* at the protein level.

#### 4.1.4. MOS Protects Tight Junction Proteins

As important members of AJCs, *ZO-1* and Occludin are two important tight junction proteins which play a key role in maintaining the connection between intestinal epithelial cell [59]. Previous studies in our lab proved that *A. hydrophila* challenge could cause down-regulation of TJs and AJs gene levels [27]. TEM results in this study also found that the gaps between intestinal epithelial cells in the *A. hydrophila* challenged group were increased, which indicated that *A. hydrophila* destroyed AJCs (Figure 1). Our results also showed that compared with the control group, the IOD of Occludin and *ZO-1* in the MOS group (400 mg/kg) were significantly increased in the three intestinal segments (*p* > 0.05), indicating that dietary MOS can provide effective protection for the tight junction protein under the condition of *A. hydrophila* infection (Figure 2 and Figure 3). It is worth noting that when excessive MOS (1000 mg/kg) was added, the IOD difference of Occludin in the MI, *ZO-1* in the PI and MI showed no significant change. Therefore, we believe that the protective effect of MOS is closely related to its supplemental level. We speculate that the reason for this difference may be partly related to intestinal digestive function, which is closely related to gut motility and chyme viscosity [7,60]. MOS is a kind of FODMAP (fermentable oligosaccharides, disaccharides, monosaccharides, and polyols) [7]. Many studies have shown that adding high doses of FODMAPs to the diet can exacerbate diarrhea, while adding low doses can significantly reduce diarrhea symptoms [7,61,62]. However, intestinal diarrhea is closely related to gut motility. Another study on rat gut showed that SCFAs could enhance gut motility, reduce intestinal chyme pass time and promote the process of evacuation [5,63]. Our previous study on grass carp intestine also confirmed that MOS supplementation increased the concentration of short-chain fatty acids (SCFAs) [30]. It has been reported in European sea bass intestine that MOS supplementation could increase the number of goblet cells [23], which can secrete mucus to protect the intestinal epithelium, lubricate the chyme and guarantee that the process of evacuation proceeds smoothly [64]. On the other hand, MOS also is a water-soluble fiber, which can swell the gastrointestinal tract and increase the viscosity of the digesta in mice [7]. This leads to an increase in satiety and a decrease in the intake absorption of nutrients in the gut. Our early growth performance data also showed that the 1000 mg/kg MOS group experience a decline in comparison with the 400 mg/kg MOS group [30]. Another study on zebrafish also found that excess MOS can decrease growth performance [65]. In summary of the above evidence, we believe that the feed of the high-dose MOS group was incompletely digested and absorbed, and that MOS was excreted from the gut directly without effective utilization. Therefore, the protective intestinal tight junctions’ effect is better with 400 mg/kg MOS than 1000 mg/kg MOS under *A. hydrophila* challenge conditions. Although we used some reliable phenotypic indicators (ultrastructural observations, biomarkers, and immunohistochemistry), they were still not sufficient to understand the mechanisms involved in the protection of intestinal structural integrity by MOS. To further investigate these protective mechanisms, we conducted systematic studies on the gene and protein levels of intestinal AJCs (tight junctions and adhesion junctions).

### 4.2. MOS Supplementation Enhanced Intestinal Structural Integrity and Possible Mechanism in Fish

#### 4.2.1. MOS Positively Regulates TJs and AJs

It is well-known that TJs and AJs are prerequisites for maintaining intestinal structural integrity [59]. In this study, by measuring the gene expression levels of a large number of TJs and AJs, it was found that dietary MOS supplementation could significantly improve the expression of the studied tight junction and adhesive junction proteins (except *ZO-2* and *Claudin-15b* in three segments) under *A. hydrophila* challenged conditions. These results indicated that MOS had a beneficial protective effect on the intestinal epithelial AJCs, which was consistent with our phenotypic results (TEM and IHC). Although there have been no other reports on the effect of MOS on intestinal AJCs in fish, similar results have been obtained in terrestrial animal studies, which showed that MOS could up-regulate *ZO-1* and *Claudin-1* in mice and up-regulated *Occludin* in broilers [66,67]. Therefore, we believe that the prebiotic effect of MOS on the animal intestinal tract may be effective across species. Interestingly, we found that MOS had no significant effect on *ZO-2* and *Claudin-15b* in the three intestinal segments of the intestine. However, this specific mechanism remains unclear and needs further study. These results indicated that dietary MOS supplementation can significantly improve TJs and AJs under *A. hydrophila* challenge, which is also important evidence for protecting the intestinal epithelial barrier. In general, TJs are mainly regulated by MLCK, while AJs are closely related to RhoA. Therefore, we will further explore the key signal molecules that regulate AJCs.

#### 4.2.2. MOS Is Involved in the Regulation of Key Signal Molecules of AJCs

According to the available literature, activating MLCK expression in mature monolayers causes MLC phosphorylation and is sufficient to cause increases in TJ permeability, and this change usually leads to a reorganization and down-regulation of TJ in Caco-2 Cell [37,68]. Another study in HT-29, DLD-1, and Caco-2 cells showed that activation of the RhoA-Rock signaling pathway can promote F-actin abnormal contraction and thus lead to down-regulation of adhesion junctions [69]. A previous study in our lab has shown similar results in grass carp intestine, that *A. hydrophila* could activate RhoA-Rock signaling and down-regulate tight junction and adhesion junctions [27]. The current study showed that dietary MOS supplementation could significantly inhibit the key signal molecules’ gene expression of MLCK and RhoA and related downstream molecules under *A. hydrophila* infection. These results suggest that MOS may improve the expression of most AJCs by participating in the regulation and inhibition of MLCK and RhoA signals simultaneously. In order to further confirm this, we also investigated the protein level of RhoA and found that it was consistent with the result of gene expression, confirming that MOS has a negative effect on RhoA. In addition, to our knowledge, LPS and ROS have been shown to directly activate MLCK and RhoA signaling in several studies in mice [37,70]. Coincidentally, MOS has also been shown to eliminate excessive ROS and compete for LPS-binding sites in vitro [42,71]. Therefore, we confirmed that MOS is involved in the regulation of MLCK and RhoA partly in this way (Figure 7). However, whether MOS can directly affect RhoA or MLCK has not been reported, and is worthy of further study in future work.

### 4.3. Correlation between AJCs and Antioxidant Capacity in Animal Intestine

Our correlation results found a highly positive correlation between the T-AOC and studied AJCs genes in the three intestinal segments of grass carp. Similar results were obtained from mirror carp (*Cyprinus carpio*) and channel catfish (*Ictalurus punctatus*) with *Yucca schidigera* extract and *Eucommia ulmoides* leaf extract supplementation [72,73]. In broiler chickens and *Caenorhabditis elegans*, it has been reported that the intestinal antioxidant capacity was significantly enhanced when probiotics (*Bacillus amyloliquefaciens* and *Lactobacillus plantarum*) were added to the diet, along with the enhancement of TJs and AJs [74,75]. This evidence suggests that there seems to be a subtle regulatory mechanism between intestinal antioxidant capacity and AJCs, but unfortunately ths mechanism has not been reported. Interestingly, these natural antioxidants and probiotics all had a significant effect on the tight junction or gut microbes [73,74,76,77,78,79]. Similarly, the effects of MOS on animal gut microbes (Japanese quail and mice) have been widely reported [80,81,82]. All of above evidence suggests that MOS can optimize gut microbes and their metabolites. Our previous study also confirmed that MOS can significantly increase the number of several probiotics (*Lactobacillus* and *Bifidobacterium*) through traditional microbial culture methods, and can also increase the content of their metabolites short-chain fatty acids (propionate and butyrate) in grass carp [30]. In conclusion, we believe that the synergistic effect of antioxidant capacity and AJCs may be related to intestinal microbes and their metabolites, but there are still some limitations to the current study. Therefore, this is an interesting direction in the future, and we look forward to further exploring these mechanisms through 16S rRNA or metagenomic and metabolomic bioinformatics analyses.

### 4.4. The Impact of MOS on Other Species’ Intestinal Structural Integrity

There exist many studies on prebiotics and their impact on the intestinal immune system of terrestrial and aquatic animals, whereas there are few studies on intestinal structural integrity. As mentioned above, only two studies reported that MOS can improve the AJCs (such as *Claudin-3* and *E-cadherin*) in fish (European sea bass and Rainbow Trout) [23,24]. In terrestrial animals, studies on broilers showed that MOS could reduce the concentration of serum DAO and endotoxin, and maintain intestinal *Occludin* and *Claudin-3* expression under enteritis model conditions [66,83]. Studies on mice ileum and Caco-2 cells displayed that MOS could up-regulate the expression of *Claudin-1*, *ZO-1* and *MUC-2* under LPS-induced conditions [67]. It has been reported that MOS could up-regulate *ZO-1*, Occludin and *Claudin-1* (not jejunum) gene expression in jejunum and duodenum of weaned piglets [84]. In addition, there is a report that exhibited that MOS could enhance tight junctions by promoting transepithelial electrical resistance (TEER) in T-84 cells [85]. From the above literature, we can find that MOS has an excellent beneficial effect on the intestinal structural integrity of most species. However, studies on the intestinal structural integrity of MOS in humans, terrestrial and aquatic animals are not comprehensive enough. The current study not only verified the phenotypic destruction of intestinal structure in animals under pathogenic bacteria challenged conditions, but also systematically the tight junction and adherent junction, and further revealed the potential pathways involved, providing a valuable reference for subsequent studies.

## 5. Conclusions

In summary, adding prebiotics to diets can contribute to improving fish intestinal health and increasing disease resistance, which is of great significance for modern intensive aquaculture. The present study improves our knowledge regarding the implementation of MOS as a prebiotic in the freshwater fish diet. Our results showed that dietary MOS supplementation can effectively protect fish intestinal structural integrity under pathogen-challenge conditions by protecting microvilli, reducing intestinal permeability, improving intestinal antioxidant capacity, and enhancing the expression of tight junctions and adherent junctions. Moreover, we further revealed that MOS effectively enhanced the tight junction and adherent junction through the inhibition of MLCK and RhoA/ROCK signalling pathways in the fish intestine. Notably, in our detailed discussion of the association between MOS excess effects and the AJCs and antioxidant activity, evidence suggest that these appear to be closely related to gut microbes. Therefore, in the future, further research is necessary on the specific mechanism of gut microbes regulating intestinal structural integrity through 16S rRNA or metagenomic and metabolomic bioinformatics analyses.

## Figures and Tables

**Figure 1 antioxidants-11-00806-f001:**
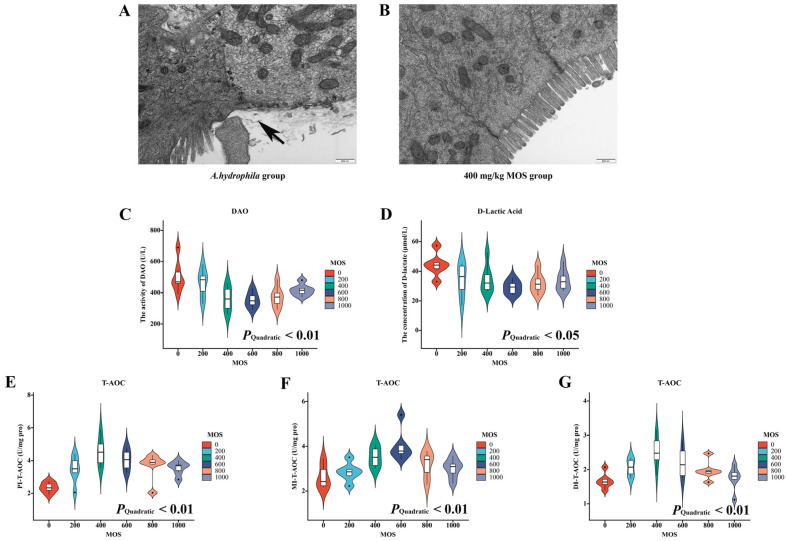
The effect of MOS on phenotypic indicators of intestinal structural integrity after infection with *Aeromonas hydrophila*. (**A**,**B**) Ultrastructural observation of intestine; (**C**,**D**) intestinal permeability parameters; (**E**–**G**) intestinal total antioxidant capacity in the PI, MI and DI. DAO: diamine oxidase (U/L); d-lactate: malondialdehyde (μmol/L); T-AOC: total antioxidant capacity; PI: proximal intestine; MI: middle intestine; DI: distal intestine. N = 6 for each MOS level, *p*-values indicate a significant quadratic dose–response relationship (*p* < 0.05).

**Figure 2 antioxidants-11-00806-f002:**
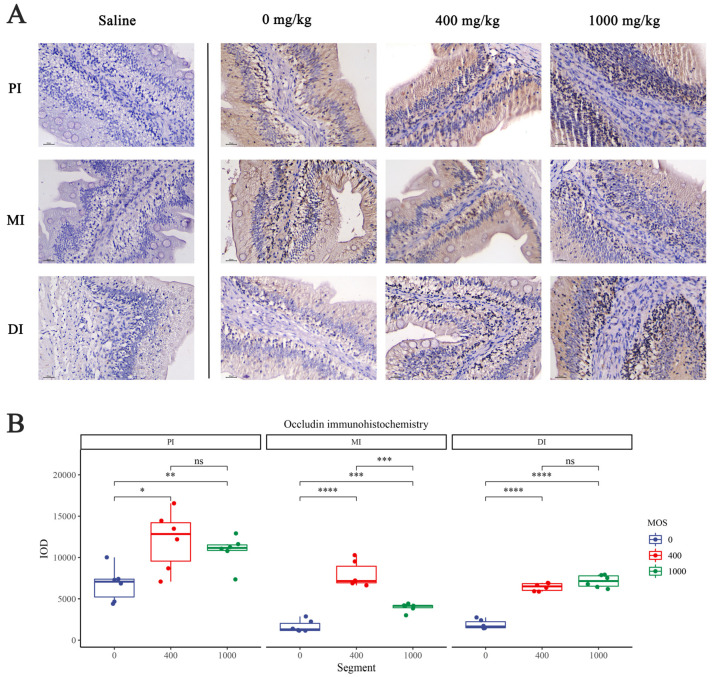
The effect of MOS on Occludin expression with the immunohistochemistry method in three intestinal segments after infection with *Aeromonas hydrophila*. (**A**) Occludin protein expression in the intestine in the 0 mg/kg MOS, 400 mg/kg MOS and 1000 mg/kg MOS groups; (**B**) Quantification of the positive area as revealed by Image Pro Plus 6.0. N = 6 for each MOS level. Differences among the variables were assessed using Student’s *t*-tests. Statistical significance: * *p* < 0.05; ** *p* < 0.01, *** *p* < 0.001; **** *p* < 0.0001; ns: not significant.

**Figure 3 antioxidants-11-00806-f003:**
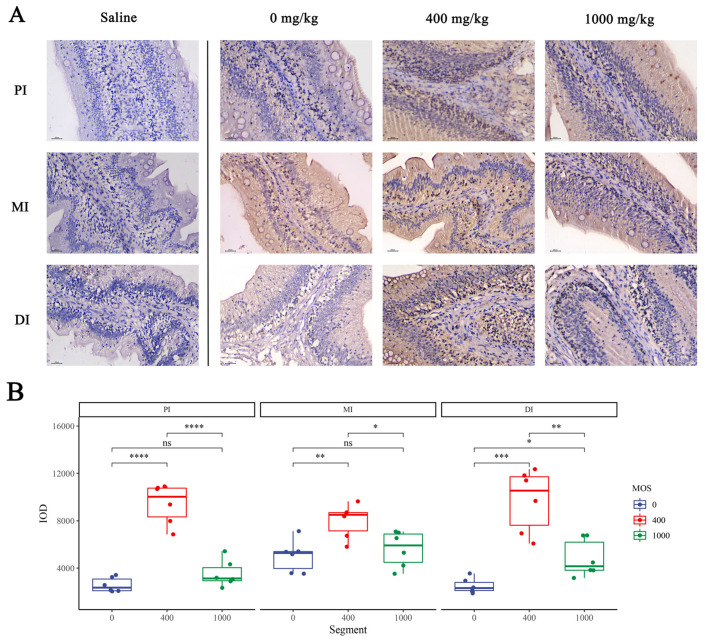
The effect of MOS on *ZO-1* expression by the immunohistochemistry method in three intestinal segments after infection with *Aeromonas hydrophila.* (**A**) *ZO-1* protein expression of intestine in the 0 mg/kg MOS, 400 mg/kg MOS and 1000 mg/kg MOS groups; (**B**) Quantification of the positive area as revealed by Image Pro Plus 6.0. N = 6 for each MOS level. Differences among the variables were assessed using Student’s *t*-tests. Statistical significance: * *p* < 0.05; ** *p* < 0.01, *** *p* < 0.001; **** *p* < 0.0001; ns: not significant.

**Figure 4 antioxidants-11-00806-f004:**
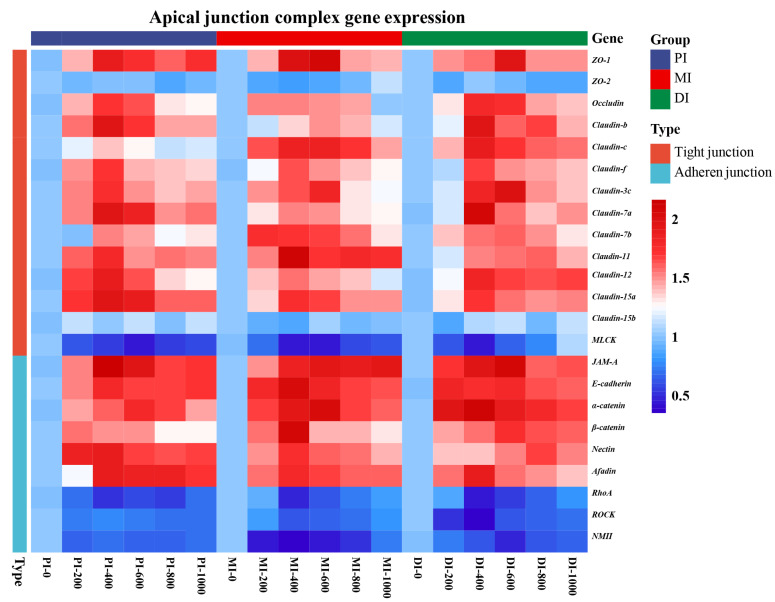
Heat map showing MOS (mg/kg diet) changed expression of AJCs and related signalling molecules genes in three intestinal segments of grass carp after infection with *Aeromonas hydrophila*. The signal values of up-regulation (red) and down-regulation (blue) were expressed and ranged from 0.5 to 2 fold. Data represent means of six fish in each group (N = 6).

**Figure 5 antioxidants-11-00806-f005:**
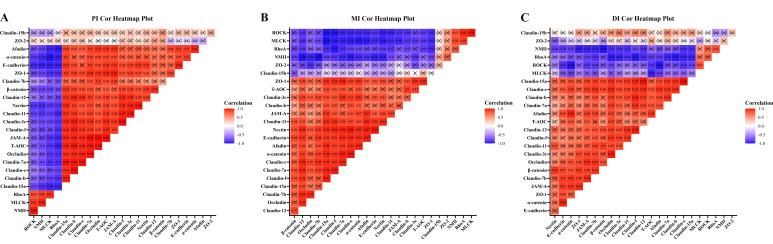
Correlation analysis of parameters in the three intestinal segments of grass carp after infection of *Aeromonas hydrophila.* (**A**) proximal intestine; (**B**) middle intestine; (**C**) distal intestine. *R* > 0.7, strong correlation; 0.5 < *R* < 0.7, moderate correlation; *R* < 0.5, weak correlation.

**Figure 6 antioxidants-11-00806-f006:**
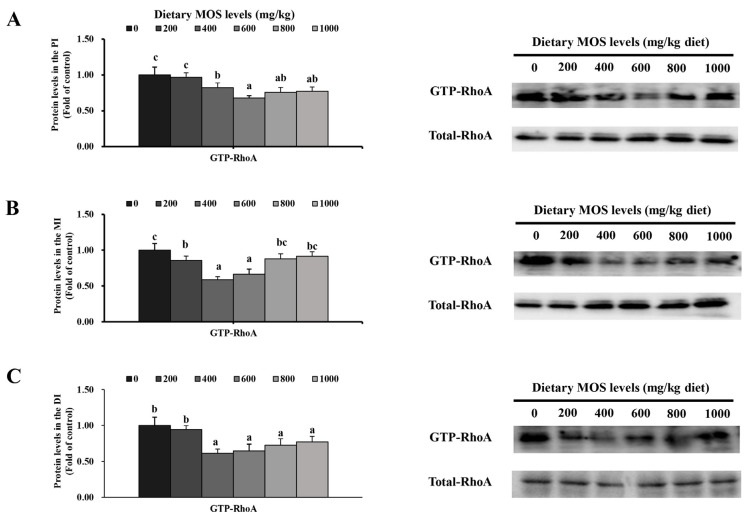
Western blot analysis of GTP-RhoA levels in the intestine of grass carp after infection with *Aeromonas hydrophila*. (**A**) Proximal intestine; (**B**) middle intestine; (**C**) distal intestine. Data represent means of three fish in each group, error bars indicate S.D. Values with different letters are significantly different (*p* < 0.05). Quantification and analysis were performed through NIH Image J software (version 1.42 *q*, National Institutes of Health, Bethesda, MD, USA).

**Figure 7 antioxidants-11-00806-f007:**
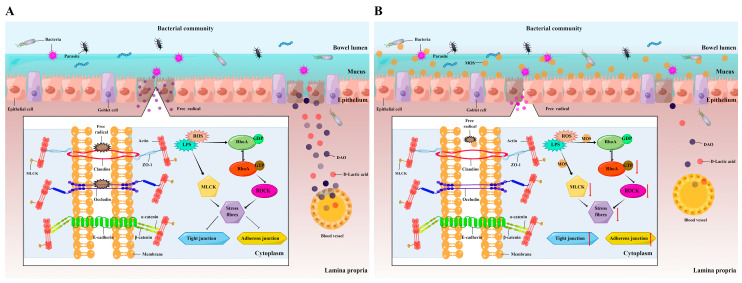
Potential action pathways of MOS on intestinal structural integrity and its related mechanisms after infection with *Aeromonas hydrophila*. (**A**) *A. hydrophila* group; (**B**) optimal MOS group. This picture is drawn by Figdraw (www.figdraw.com, accessed on 24 March 2022).

## Data Availability

The data presented in this study are available in review.

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
