# Peer review of "An Antioxidant Supplement Function Exploration: Rescue of Intestinal Structure Injury by Mannan Oligosaccharides after Aeromonas hydrophila Infection in Grass Carp (Ctenopharyngodon idella)"

_antioxidants, 2022, doi:10.3390/antiox11050806_

Round 1

Reviewer 1 Report

This is a meticulously constructed experiment with the objective of determining the beneficial effect of dietary MOS supplementation on the function and integrity of the fish gut.

A minor revision to the introduction and a significant revision to the conclusion would improve the manuscript.

For instance, I would anticipate beginning the introduction with references to the benefits of pre and probiotics in animal and fish feeds then gradually moving on to the specifics of grass carp aquaculture.

I'm also curious about the fish's gut microbiome at the start of the experiment; this could have an effect on the outcome.

These are some specific recommendations for improving the manuscript.

Avoid big statements such as these in the Abstract.

Line 133: I propose you exclude the term "re-fixed" and instead use the term "post-fixed in 1% osmium tetroxide."

Lines 32-33. Present the results of each marker you utilized and the conclusions that may be drawn from them, for example, what you observed about intestinal integrity, feed conversion, disease prophylaxis, and other assays/indicators.

Line 226 The reader will be perplexed; what exactly do you mean by "appropriate response"?

Lines 484-489 This is your first mention of probiotics. You should include a paragraph about probiotics in the beginning; there are several recent publications that you can cite and indicate the importane of intestinal functional integrity and the significance of probiotic bacteria for feed conversion and growth.

Lines 511-523.  To avoid confusing the reader, you should prioritize the points to conclude, beginning with the most significant benefits observed and gradually adding collaborative evidence with the parameters that support your conclusion, and finally, on the basis of some of your findings, propose your own ideas for future research.

These are some recently published papers to consider in your revision.

Gao, Y., Huo, X., Wang, Z., Yuan, G., Liu, X., Ai, T., & Su, J. (2021). Oral Administration of Bacillus subtilis Subunit Vaccine Significantly Enhances the Immune Protection of Grass Carp against GCRV-II Infection. Viruses, 14(1), 30.

Nathanailides, C., Kolygas, M., Choremi, K., Mavraganis, T., Gouva, E., Vidalis, K., & Athanassopoulou, F. (2021). Probiotics Have the Potential to Significantly Mitigate the Environmental Impact of Freshwater Fish Farms. Fishes, 6(4), 76.

Iorizzo, M., Albanese, G., Letizia, F., Testa, B., Tremonte, P., Vergalito, F., ... & Sorrentino, E. (2022). Probiotic Potentiality from Versatile Lactiplantibacillus plantarum Strains as Resource to Enhance Freshwater Fish Health. Microorganisms, 10(2), 463

Author Response

Response to Reviewer 1 Comments

 General Comment: This is a meticulously constructed experiment with the objective of determining the beneficial effect of dietary MOS supplementation on the function and integrity of the fish gut.A minor revision to the introduction and a significant revision to the conclusion would improve the manuscript. For instance, I would anticipate beginning the introduction with references to the benefits of pre and probiotics in animal and fish feeds then gradually moving on to the specifics of grass carp aquaculture. I'm also curious about the fish's gut microbiome at the start of the experiment; this could have an effect on the outcome.

Response : Thanks for your kind suggestions and comments, which are all valuable and very helpful for revising and improving our paper, as well as the important guiding significance to our research. We have studied comments carefully and have made correction in Introduction section and Conclusion section which we hope meet with approval.

Thank you again for your comments!

Point 1: Avoid big statements such as these in the Abstract.

Point 3: Lines 32-33. Present the results of each marker you utilized and the conclusions that may be drawn from them, for example, what you observed about intestinal integrity, feed conversion, disease prophylaxis, and other assays/indicators.

Response 1 and 3: Thank you for your kind comments. We carefully rewrote the Abstract section according to your helpful suggestion. Your suggestion is very good. The detailed changes are as follows:

Lines 22-39:

Abstract: Mannan oligosaccharides (MOS) is a sort of functional oligosaccharides, which has re-ceived more attention because of its beneficial effects on fish intestinal health. However, intestinal structural integrity is a necessary prerequisite for intestinal health. This study focused on exploring the protective effects of dietary MOS supplementation on the grass carp (Ctenopharyngodon idella) intestinal structural integrity (include tight junction (TJ) and adherent junction (AJ)) and its related signalling molecule mechanism. A total of 540 grass carp (215.85 ± 0.30 g) were fed six diets con-taining graded levels of dietary MOS supplementation (0, 200, 400, 600, 800 and 1000 mg/kg) for 60 days. Subsequently, a challenge test was conducted by injection of Aeromonas hydrophila for 14 days. We used ELISA, spectrophotometry, transmission electron microscope, immunohistochemistry, qRT-PCR and Western blotting to determine the effect of dietary MOS supplementation on intes-tinal structural integrity and antioxidant capacity. The results revealed that dietary MOS sup-plementation protected the microvillus of intestine; reduced serum diamine oxidase and D-lactate levels (P< 0.05); enhanced intestinal total antioxidant capacity (P< 0.01); up-regulated intestinal most of TJ and AJ mRNA levels and decreased GTP-RhoA protein levels (P< 0.01). In addition, we also found several interesting results that MOS supplementation has no effects on ZO-2 and Claudin-15b. Overall, these findings suggested that dietary MOS supplementation could protect intestinal ultrastructure, reduce intestinal mucosal permeability and maintain intestinal structural integrity via inhibiting MLCK and RhoA/ROCK signalling pathways.

We hope our answer can get your agreement and recognition. If you have any questions, please do not hesitate to contact me. Thank you again for your comments!

Point 2: Line 133: I propose you exclude the term "re-fixed" and instead use the term "post-fixed in 1% osmium tetroxide.".

Response 2: Thank you very much for your kind comments. We are very sorry for the inaccuracy of the description in the Method section. According to your helpful suggestions, we have revised the description:

Lines 151-154:

The sentence “Samples from different groups and different intestinal segments were quickly fixed in 2.5% glutaraldehyde in cacodylate buffer (Lilai Biotechnology, China) after collection and followed were re-fixed in 1% osmium tetroxide (Leica, Germany).”

has been revised to

“Samples from different groups and different intestinal segments were quickly fixed in 2.5% glutaraldehyde in cacodylate buffer (Lilai Biotechnology, China) after collection and followed were post-fixed in 1% osmium tetroxide (Leica, Germany).”

Thank you again for your comments!

Point 4: Line 226 The reader will be perplexed; what exactly do you mean by "appropriate response"?

Response 4: Thank you very much for your kind comment and reminder. We are so sorry for the inaccurate writing in our manuscript. What we mean by that is the “optimal response”. According to your helpful suggestions, we have revised the description:

Lines 255-257:

The sentence “where the MOS group (400 mg/kg) has exhibited the appropriate response on antioxidant capacity in the PI and DI, and 600 mg/kg MOS in the MI.”

has been revised to

“where the MOS group (400 mg/kg) has exhibited the optimal response on antioxidant capacity in the PI and DI, and 600 mg/kg MOS in the MI.”

Thank you again for your comments!

Point 5: Lines 484-489 This is your first mention of probiotics. You should include a paragraph about probiotics in the beginning; there are several recent publications that you can cite and indicate the importane of intestinal functional integrity and the significance of probiotic bacteria for feed conversion and growth.

These are some recently published papers to consider in your revision.

Gao, Y., Huo, X., Wang, Z., Yuan, G., Liu, X., Ai, T., & Su, J. (2021). Oral Administration of Bacillus subtilis Subunit Vaccine Significantly Enhances the Immune Protection of Grass Carp against GCRV-II Infection. Viruses, 14(1), 30.

Nathanailides, C., Kolygas, M., Choremi, K., Mavraganis, T., Gouva, E., Vidalis, K., & Athanassopoulou, F. (2021). Probiotics Have the Potential to Significantly Mitigate the Environmental Impact of Freshwater Fish Farms. Fishes, 6(4), 76.

Iorizzo, M., Albanese, G., Letizia, F., Testa, B., Tremonte, P., Vergalito, F., ... & Sorrentino, E. (2022). Probiotic Potentiality from Versatile Lactiplantibacillus plantarum Strains as Resource to Enhance Freshwater Fish Health. Microorganisms, 10(2), 463.

Response 5: Thanks for your kind suggestions and comments, which are all valuable and very helpful for revising and improving our paper, as well as the important guiding significance to our research. Your suggestion is very good. We also agree with you that probiotics and prebiotics and their relationship should be discussed in the beginning (Introduction section). This revision will make our manuscript logically clearer. Thank you very much for your recommend references, which are very helpful to our manuscript. According to your hints, our specific modifications are as follows:

Lines 42-55:

Added descriptions of probiotics: “ According to the Food and Agriculture Organization/World Health Organization (FAO/WHO) definition, probiotics are ‘live microorganisms that, when administered in adequate amounts, confer a health benefit on the host [1]. Many studies indicate that probiotics (such as Bacillus sp., Lactobacillus sp., Enterococcus sp., and so on) contribute to improving fish gut health and structural integrity [2-6]. Based on the available literature, probiotics play a beneficial role by optimizing the intestinal flora, improving the apparent digestibility of feed, and fermentation of water-soluble fiber (such as oligosaccharides) in food, and so on [3,7-9]. In addition, the beneficial effects of short-chain fatty acids (acetic, propionic and butyric), probiotic metabolites, on intestinal structure and function have been reported [10,11]. Prebiotics is like nutriment for several beneficial bacteria and are usually not affected by digestive enzymes in the gut, but can be used by probiotics to promote their growth [12]. Therefore, the application of prebiotics in aquaculture is considered an effective solution to improve feed conversion efficiency and promote fish growth [13]. ”

We hope our answer can get your agreement and recognition. If you have any questions, please do not hesitate to contact me. Thank you again for your comments!

Added References

  1. Probiotics in food : health and nutritional properties and guidelines for evaluation : Report of a Joint FAO/WHO Expert Consultation on Evaluation of Health and Nutritional Properties of Probiotics in Food including Powder Milk with Live Lactic Acid Bacteria, Cordoba, Argentina, 1-4 October 2001 [and] Report of a Joint FAO/WHO Working Group on Drafting Guidelines for the Evaluation of Probiotics in Food, London, Ontario, Canada, 30 April -1 May 2002; Food and Agriculture Organization of the United Nations, World Health Organization: Rome [Italy], 2006.
  2. Islam, S.M.M.; Rohani, M.F.; Shahjahan, M. Probiotic yeast enhances growth performance of Nile tilapia (Oreochromis niloticus) through morphological modifications of intestine. Aquaculture Reports 2021, 21, doi:10.1016/j.aqrep.2021.100800.
  3. Nathanailides, C.; Kolygas, M.; Choremi, K.; Mavraganis, T.; Gouva, E.; Vidalis, K.; Athanassopoulou, F. Probiotics Have the Potential to Significantly Mitigate the Environmental Impact of Freshwater Fish Farms. Fishes 2021, 6, doi:10.3390/fishes6040076.
  4. Iorizzo, M.; Albanese, G.; Letizia, F.; Testa, B.; Tremonte, P.; Vergalito, F.; Lombardi, S.J.; Succi, M.; Coppola, R.; Sorrentino, E. Probiotic Potentiality from Versatile Lactiplantibacillus plantarum Strains as Resource to Enhance Freshwater Fish Health. Microorganisms 2022, 10, doi:10.3390/microorganisms10020463.
  5. Cheng, J.; Laitila, A.; Ouwehand, A.C. Bifidobacterium animalis subsp. lactis HN019 Effects on Gut Health: A Review. Front Nutr 2021, 8, 790561, doi:10.3389/fnut.2021.790561.
  6. Gao, Y.; Huo, X.; Wang, Z.; Yuan, G.; Liu, X.; Ai, T.; Su, J. Oral Administration of Bacillus subtilis Subunit Vaccine Significantly Enhances the Immune Protection of Grass Carp against GCRV-II Infection. Viruses 2021, 14, doi:10.3390/v14010030.
  7. Hanau, S.; Almugadam, S.H.; Sapienza, E.; Cacciari, B.; Manfrinato, M.C.; Trentini, A.; Kennedy, J.F. Schematic overview of oligosaccharides, with survey on their major physiological effects and a focus on milk ones. Carbohydrate Polymer Technologies and Applications 2020, 1, doi:10.1016/j.carpta.2020.100013.
  8. Hill, C.; Guarner, F.; Reid, G.; Gibson, G.R.; Merenstein, D.J.; Pot, B.; Morelli, L.; Canani, R.B.; Flint, H.J.; Salminen, S.; et al. Expert consensus document. The International Scientific Association for Probiotics and Prebiotics consensus statement on the scope and appropriate use of the term probiotic. Nat Rev Gastroenterol Hepatol 2014, 11, 506-514, doi:10.1038/nrgastro.2014.66.
  9. Flint, H.J.; Scott, K.P.; Louis, P.; Duncan, S.H. The role of the gut microbiota in nutrition and health. Nat Rev Gastroenterol Hepatol 2012, 9, 577-589, doi:10.1038/nrgastro.2012.156.
  10. Deleu, S.; Machiels, K.; Raes, J.; Verbeke, K.; Vermeire, S. Short chain fatty acids and its producing organisms: An overlooked therapy for IBD? EBioMedicine 2021, 66, 103293, doi:10.1016/j.ebiom.2021.103293.
  11. Parada Venegas, D.; De la Fuente, M.K.; Landskron, G.; Gonzalez, M.J.; Quera, R.; Dijkstra, G.; Harmsen, H.J.M.; Faber, K.N.; Hermoso, M.A. Short Chain Fatty Acids (SCFAs)-Mediated Gut Epithelial and Immune Regulation and Its Relevance for Inflammatory Bowel Diseases. Frontiers in immunology 2019, 10, 277, doi:10.3389/fimmu.2019.00277.
  12. Green, M.; Arora, K.; Prakash, S. Microbial Medicine: Prebiotic and Probiotic Functional Foods to Target Obesity and Metabolic Syndrome. International journal of molecular sciences 2020, 21, doi:10.3390/ijms21082890.
  13. Lauzon, H.L.; Dimitroglou, A.; Merrifield, D.L.; Ringø, E.; Davies, S.J. Probiotics and Prebiotics: Concepts, Definitions and History. In Aquaculture Nutrition; 2014; pp. 169-184, doi:10.1002/9781118897263.ch7.

Point 6: Lines 511-523.  To avoid confusing the reader, you should prioritize the points to conclude, beginning with the most significant benefits observed and gradually adding collaborative evidence with the parameters that support your conclusion, and finally, on the basis of some of your findings, propose your own ideas for future research.

Response 6: Thank you very much for your kind comment. We are so sorry for that we don’t make it clear. According to your kind suggestion, we have rewritten the Conclusion section as follows.

Line 579-594:

Conclusion: In summary, adding prebiotics to diets can contribute to improving fish intestinal health and increase disease resistance, which is of great significance for modern intensive aquaculture. The present study improves the application knowledge for the implementation of MOS as a prebiotic in the freshwater fish diet. Our results showed that dietary MOS supplementation can effectively protect fish intestinal structural integrity under pathogen challenge conditions by protecting microvilli, reducing intestinal permeability, improving intestinal antioxidant capacity, and enhancing the expression of tight junctions and adherent junctions. Moreover, we further revealed that MOS effectively enhanced tight junc-tion and adherent junction through the inhibition of MLCK and RhoA/ROCK signalling pathways in the fish intestine. Notably, in our detailed discussion of the association between MOS excess effects and the AJCs and antioxidant activity, multiple evidence suggest that these appear to be closely related to gut microbes. Therefore, in the future, further research is necessary on the specific mechanism of gut microbes regulating intestinal structural integrity through the 16S rRNA or metagenomics and metabolomics bioinformatics analysis.

We hope our answer can get your agreement and recognition. If you have any questions, please do not hesitate to contact me. Thank you again for your comments!

Reviewer 2 Report

This study focused on exploring the protective effects of dietary MOS supplementation on the grass carp (Ctenopharyngodon idella) intestinal apical junctional complex (AJC) (include tight junction (TJ) and adherent junction (AJ)) and its related signalling  molecule mechanism. The study of functional feed additives in aquaculture is an important issue

Title: Novel antioxidant supplement function exploration: rescue of intestinal structure injury by mannan oligosaccharides after Aeromonas hydrophila infection in grass carp (Ctenopharyngodon idella).  Please adjust the title. Remove the word "novel"

Abstract:

Please highlight the main methodology.

Keywords:

Mannan oligosaccharides and Grass carp. These words are in the title of the manuscript. Please replace.

 Introduction:

Lines 45 to 46 “Mannan oligosaccharides (MOS), a novel antioxidant supplement, are widely used in aquaculture because they promote growth”. The specific mechanism of MOS on structural integrity of intestine has not been systemically investigated but it is not a novel antioxidant supplement. Please replace and adjust.

Materials and methods:

Lines 94  “The present study used the same growth trial as that of our previous study [17].”  Please describe methodology. It is a continuation of the previous work but a description of the methodology must be provided.

Lines 111 “with natural light conditions.”. Natural Light condition must be described.

Line 123 “centrifugation (3000 r/min) for 10 minutes”. Please, express as g.

Line 184 “2.9. Western blotting” same of 2.8. Statistical analyses?

Results:

Line 211 “a significant linear or quadratic dose response relationship (P < 0.05).” There is no linear response and P< 0.01 is observed.

Line 215 “Figure 1B-C.” This does not match the text

Line 223 “Figure 1D-F” This does not match the text

Line 226 “appropriate response” What does it mean? Is it the best result?

Line 300 to 301 “Figure 5. Correlation analysis of parameters in the three intestinal segments of grass carp after infection of Aeromonas hydrophila. (A) proximal intestine; (B) middle intestine; (C) distal intestine.” Statistical significance is necessary in figure legend  

Discussion:

Line 346 “Generally, MOS, a classical prebiotic, derived from Saccharomyces cerevisiae cell 346 wall [29].” Incomplete.

Conclusion:

Lines 508 to 511 “In summary, in Figure 7, our data exhibited dietary MOS supplementation (200-800 5mg kg-1) could promote intestinal health under A. hydrophila challenged, reflecting in improving protect intestinal structural integrity and enhancing intestinal antioxidant capacity of grass carp” These are results and not conclusions. Please remove figure 7

Author Response

Response to Reviewer 2 Comments

General Comment: This study focused on exploring the protective effects of dietary MOS supplementation on the grass carp (Ctenopharyngodon idella) intestinal apical junctional complex (AJC) (include tight junction (TJ) and adherent junction (AJ)) and its related signalling molecule mechanism. The study of functional feed additives in aquaculture is an important issue.

Response: Thanks for your kind suggestions and comments, which are all valuable and very helpful for revising and improving our paper, as well as the important guiding significance to our researches. We have studied comments carefully and have made correction which we hope meet with approval.

Point 1: Title:

Novel antioxidant supplement function exploration: rescue of intestinal structure injury by mannan oligosaccharides after Aeromonas hydrophila infection in grass carp (Ctenopharyngodon idella). Please adjust the title. Remove the word "novel".

Response 1: Thank you very much for your kind comment and reminding. We are so sorry for our inaccurate writing in our manuscript. According to your helpful suggestions, we have corrected the description:

The Title “Novel antioxidant supplement function exploration: rescue of intestinal structure injury by mannan oligosaccharides after Aeromonas hydrophila infection in grass carp (Ctenopharyngodon idella)”

has been revised to

“An antioxidant supplement function exploration: rescue of intestinal structure injury by mannan oligosaccharides after Aeromonas hydrophila infection in grass carp (Ctenopharyngodon idella)”

Thank you again for your comments!

Point 2: Abstract:

Please highlight the main methodology.

Response 2: Thank you very much for your kind comment and reminding. We are sorry for neglecting description of methodology in Abstract section. According to your kind suggestion, we have added the methodology in Abstract section as follows.

Line 27 to32:

The sentence “A total of 540 grass carp (215.85 ± 0.30 g) were fed six diets containing graded levels of dietary MOS supplementation (0, 200, 400, 600, 800 and 1000 mg/kg) for 60 days. Subsequently, a challenge test was conducted by injection of Aeromonas hydrophila for 14 days. We used ELISA, spectrophotometry, transmission electron microscope, immunohistochemistry, qRT-PCR and Western blotting to determine the effect of dietary MOS supplementation on intestinal structural integrity and antioxidant capacity.” has been added.

Thank you again for your comments!

Point 3: Keywords:

Mannan oligosaccharides and Grass carp. These words are in the title of the manuscript. Please replace.

Response 3: Many thanks for your kind words and your useful suggestions. We have revised the Keywords according to your suggestion.

Lines 40:

The Keywords “Mannan oligosaccharides; Intestine; Apical junctional complex; Antioxidant capacity; Grass carp” has been revised to “Permeability; Intestine; Tight junction; Adherent junction; Antioxidant capacity”

Thank you again for your comments!

Point 4: Introduction:

Lines 45 to 46 “Mannan oligosaccharides (MOS), a novel antioxidant supplement, are widely used in aquaculture because they promote growth”. The specific mechanism of MOS on structural integrity of intestine has not been systemically investigated but it is not a novel antioxidant supplement. Please replace and adjust.

Response 4: Thank you very much for your kind comment. We are so sorry for our inaccurate writing in our manuscript. According to your helpful suggestions, we have corrected the description:

Lines 62 to 64:

“Mannan oligosaccharides (MOS), a novel antioxidant supplement, are widely used in aquaculture because they promote growth performance and functional organ health [5].”

has been revised to

“Although a common prebiotic, mannan oligosaccharides (MOS) are recommended as aquaculture additives because of their excellent antioxidant properties [18].”

Thank you again for your comments!

Point 5: Materials and methods:

  • Lines 94 “The present study used the same growth trial as that of our previous study [17].”

Please describe methodology. It is a continuation of the previous work but a description of the methodology must be provided.

  • Lines 111 “with natural light conditions.”. Natural Light condition must be described.
  • Lines 123 “centrifugation (3000 r/min) for 10 minutes”. Please, express as g.
  • Lines 184 “2.9. Western blotting” same of 2.8. Statistical analyses?

Response 5: Many thanks for your kind words and your useful comments and suggestions, which are all valuable and very helpful for revising and improving our paper, as well as the important guiding significance to our researches. Sorry about some inaccurate writing and incomplete description. We have studied your comments carefully and have made correction which we hope to meet with approval. The detail changes are as follows:

(1) Response: Lines 94 The sentence“The present study used the same growth trial as that of our previous study [17].” has been deleted. The sentence“The present study used the same growth trial as that of our previous study [17].” has been added in Line 118.

According to your helpful suggestions, we added some descriptions of methodology and rewritten 2.2 Animal section:

Lines 117 to 127:

2.2. Animal

The present study used the same growth trial as that of our previous study [30]. In brief, a total of 540 healthy grass carp are provided from Tong Wei farm (Chengdu, China). The average weight of fish is 215.85 ± 0.30 g. To adapt grass carp to the environment, one month of domestication was performed before the formal trial began. Thereafter, all fish were randomly distributed to 18 experimental cages (1.4 m L ×1.4 m W ×1.4 m H) with working volume. There were 30 fish per cage. Mesh discs (100 cm diameter) are used to assemble leftover diets per cage. The growth test lasted for 60 days (feed 4 times/day). After feeding for 30 min, leftover diets were collected, dehydrated and weighed. The current study and methods were allowed by the Animal Care Advisory Committee of Sichuan Agricultural University (permit no. LZY-2018114005).

(2) Response: Lines 132 to 133 “with natural light conditions.” has been revised to “natural light conditions (natural long day in summer: 14 h light/10 h dark).”

(3) Response: Lines 144 “centrifugation (3000 r/min) for 10 minutes.” has been revised to “centrifugation at 845g for 10 minutes”.

(4) Response: We are very sorry about this errors. Lines 204 “2.9. Western blotting” has been revised to “2.9. Statistical analysis”.

Thank you again for your comments!

Point 6: Results:

  • Line 211 “a significant linear or quadratic dose response relationship (P < 0.05).” There is no linear response and P< 0.01 is observed.
  • Line 215 “Figure 1B-C.” This does not match the text
  • Line 223 “Figure 1D-F” This does not match the text
  • Line 226 “appropriate response” What does it mean? Is it the best result?
  • Line 300 to 301 “Figure 5. Correlation analysis of parameters in the three intestinal segments of grass carp after infection of Aeromonas hydrophila. (A) proximal intestine; (B) middle intestine; (C) distal intestine.” Statistical significance is necessary in figure legend

Response 6: Many thanks for your kind words and your useful comments and notice, which are all valuable and very helpful for revising and improving our paper. Sorry about some errors. We have studied your comments carefully and have made correction which we hope to meet with approval. The detail changes are as follows:

(1) Response: Lines 241 to 242 “a significant linear or quadratic dose response relationship (P < 0.05).” has been revised to “a significant quadratic dose response relationship (P < 0.05).”

(2) Response: Lines 246 “Figure 1B-C.” has been revised to “Figure 1C-D.”

(3) Response: Lines 253 “Figure 1D-F” has been revised to “Figure 1E-G.”

(4) Response: We are so sorry for our inaccurate writing in our manuscript. What we mean by that is the “optimal response”. Lines 256“appropriate response” has been revised to “optimal response”.

(5) Response: Lines 329 Figure 5. “R>0.7, strong correlation; 0.5<R<0.7, moderate correlation; R<0.5, weak correlation“. has been added.

Thank you again for your comments!

Point 7: Discussion:

Lines 346 “Generally, MOS, a classical prebiotic, derived from Saccharomyces cerevisiae cell 346 wall [29].” Incomplete.

Response 7: Thank you very much for your kind comment and reminding. We are sorry for this incomplete sentence. According to your kind suggestion, we have revised as follows:

Lines 375 to 376

“Generally, MOS, a classical prebiotic, derived from Saccharomyces cerevisiae cell 346 wall [29].”

has been revised to “Generally, MOS, a classical prebiotic, derived from Saccharomyces cerevisiae cell wall, is widely used in animal production [41].”

Thank you again for your comments!

Point 8: Conclusion: Lines 508 to 511 “In summary, in Figure 7, our data exhibited dietary MOS supplementation (200-800 mg kg-1) could promote intestinal health under A. hydrophila challenged, reflecting in improving protect intestinal structural integrity and enhancing intestinal antioxidant capacity of grass carp” These are results and not conclusions. Please remove figure 7

Response 8: Thank you for your kind comments. We carefully rewritten the Conclusion section according to your helpful suggestion. Your suggestion is very good.

Lines 580 to 594:

In summary, adding prebiotics to diets can contribute to improving fish intestinal health and increase disease resistance, which is of great significance for modern intensive aquaculture. The present study improves the application knowledge for the implementation of MOS as a prebiotic in the freshwater fish diet. Our results showed that dietary MOS supplementation can effectively protect fish intestinal structural integrity under pathogen challenge conditions by protecting microvilli, reducing intestinal permeability, improving intestinal antioxidant capacity, and enhancing the expression of tight junctions and adherent junctions. Moreover, we further revealed that MOS effectively enhanced tight junction and adherent junction through the inhibition of MLCK and RhoA/ROCK signalling pathways in the fish intestine. Notably, in our detailed discussion of the association between MOS excess effects and the AJCs and antioxidant activity, multiple evidence suggest that these appear to be closely related to gut microbes. Therefore, in the future, further research is necessary on the specific mechanism of gut microbes regulating intestinal structural integrity through the 16S rRNA or metagenomics and metabolomics bioinformatics analysis.

We hope our answer can get your agreement and recognition. If you have any questions, please do not hesitate to contact me. Thank you again for your comments!

Reviewer 3 Report

Paper title: Novel antioxidant supplement function exploration: rescue of intestinal structure injury by mannan oligosaccharides after Aeromonas hydrophila infection in grass carp (Ctenopharyngodon idella) described by Lu et al.

Is interesting however this paper, in my opinion, is not suitable for print in the Antioxidant journal because the topic of this journal is different from than topic of this paper.

This paper doesn’t present results from any methods for antioxidant activity measurement. Authors only suggest that MOS presents potential antioxidant activity but they don’t evaluate it. Therefore I propose rejected because this paper is out of the scope of journal's aim.

Author Response

Response to Reviewer 3 Comments

General Comment: Is interesting however this paper, in my opinion, is not suitable for print in the Antioxidant journal because the topic of this journal is different from than topic of this paper. This paper doesn’t present results from any methods for antioxidant activity measurement. Authors only suggest that MOS presents potential antioxidant activity but they don’t evaluate it. Therefore I propose rejected because this paper is out of the scope of journal's aim.

Response :

Thank you very much for your review and recognition that our study is interesting. We think it is necessary to make a detailed explanation of the problem that the topic of our research you mentioned is not in line with this journal.

In this paper, we mainly focus on the mechanism of mannan oligosaccharides (MOS) on the intestinal structural integrity of grass carp under A.hydrophila challenged conditions. Our target substance MOS is not only a prebiotic but also a class of natural antioxidants, which has been confirmed in many studies [1-4]. We also found that it has an excellent antioxidant function in vitro in the previous study [5].

Therefore, we carefully read the Aims and Scope of Antioxidants and finally selected the Outcomes of Antioxidants and Oxidative Stress section (Special Issue: Antioxidants Benefits in Aquaculture). https://www.mdpi.com/journal/antioxidants/special_issues/antioxidants_aquaculture. This research comprehensively evaluated the effects of MOS on grass carp intestine structure integrity by investigating some phenotypic indexes such as intestinal ultrastructure, intestinal permeability, total antioxidant capacity, tight junction immunohistochemical expression and apical junction complexes (AJCs) related gene expression and key signaling molecule RhoA protein levels under A.hydrophila challenged condition.

Hence, the content of this study is highly related to the three topics mentioned in the Special Issue: Antioxidants Benefits in Aquaculture, which is consistent with the relevant topics (Dietary antioxidants and food supplements; Antioxidant products in intestinal health; The relationship between antioxidants and health). In addition, specific indicators related to antioxidant capacity were also carried out in this study. In this study, the total antioxidant capacity of the intestine (proximal intestine (PI), middle intestine (MI) and distal intestine (DI)) of grass carp was investigated. Further correlation analysis showed that there was a high correlation between AJCs and antioxidant capacity. Although the related mechanisms involved were not directly revealed, combined with previous studies, the synergistic effect of intestinal probiotics and their metabolites may be involved, which also provides a valuable reference for future studies. Therefore, we believe that our research is suitable for the requirements of this journal.

We hope our answer can get your agreement and recognition. Thank you again for your comments!

  1. Jana, U.K.; Kango, N. Characteristics and bioactive properties of mannooligosaccharides derived from agro-waste mannans. Int J Biol Macromol 2020, 149, 931-940, doi:10.1016/j.ijbiomac.2020.01.304.
  2. Vieira, T.F.; Corrêa, R.C.; Peralta, R.A.; Peralta-Muniz-Moreira, R.F.; Bracht, A.; Peralta, R.M. An overview of structural aspects and health beneficial effects of antioxidant oligosaccharides. Current Pharmaceutical Design 2020, 26, 1759-1777.
  3. Spring, P.; Wenk, C.; Connolly, A.; Kiers, A. A review of 733 published trials on Bio-Mos®, a mannan oligosaccharide, and Actigen®, a second generation mannose rich fraction, on farm and companion animals. Journal of Applied Animal Nutrition 2015, 3, doi:10.1017/jan.2015.6.
  4. Bland, E.J.; Keshavarz, T.; Bucke, C. The influence of small oligosaccharides on the immune system. Carbohydr Res 2004, 339, 1673-1678, doi:10.1016/j.carres.2004.05.009.
  5. Lu, Z.; Feng, L.; Jiang, W.D.; Wu, P.; Liu, Y.; Jiang, J.; Kuang, S.Y.; Tang, L.; Li, S.W.; Liu, X.A.; et al. Mannan Oligosaccharides Application: Multipath Restriction From Aeromonas hydrophila Infection in the Skin Barrier of Grass Carp (Ctenopharyngodon idella). Frontiers in immunology 2021, 12, 742107, doi:10.3389/fimmu.2021.742107.

Reviewer 4 Report

Review of “Novel antioxidant supplement function exploration: rescue of intestinal structure injury by mannan oligosaccharides after Aeromonas hydrophila infection in grass carp” (antioxidants-1672680)

This study investigated and showed that the positive effect of MOS on the promote intestinal health in grass carp and showed that this effect was mediated by maintaining ultrastructural integrity. This study was interesting. To revise the Results and Discussion sections are desirable.

Results
Are there differences in phenotypes such as life expectancy and weight by MOS supplementation?
Furthermore, in figure 2 and figure 3, this reviewer has the impression that 400 mg/kg MOS is more effective than 1000 mg/kg. Please discuss this point.

Discussion
How about the impact of MOS on other animal species?

Author Response

Response to Reviewer 4 Comments

General Comment: This study investigated and showed that the positive effect of MOS on the promote intestinal health in grass carp and showed that this effect was mediated by maintaining ultrastructural integrity. This study was interesting. To revise the Results and Discussion sections are desirable.

Response: Thanks for your kind suggestions and comments, which are all valuable and very helpful for revising and improving our paper, as well as the important guiding significance to our research. We have studied comments carefully and have made correction which we hope meet with approval.

Thank you again for your comments!

Point 1: Results:

Are there differences in phenotypes such as life expectancy and weight by MOS supplementation?

Response 1: Thank you very much for your kind comment. According to our published experimental data, after the growth trial, we found that the growth performance of the MOS supplementation group showed an obvious quadratic dose-response (P<0.01). Among them, the final body weight (FBW), percent weight gain (PWG) and specific growth rate (SGR) of grass carp in 400 mg/kg MOS group reached the maximum. After the challenge, our fish had a 100% survival rate for the entire duration of the trial because we were using a non-lethal challenge dose. According to the morbidity data, compared with the control group, the enteritis morbidity and red-skin morbidity of the optimal group (400 mg/kg MOS) were decreased by 53.45% and 42.56%, respectively. These results indicate that dietary MOS supplementation has beneficial effects on animal growth and health, and the optimal supplemental level can achieve the maximum effect.

According to your kind suggestion, we added 3.1. Growth and disease resistance phenotypes in the Results section. The detail is as follows:

Lines 215 to 226

3.1. Growth and disease resistance phenotypes

The current study used the same growth trial from our previous work on grass carp [30]. after 60 days of growth trial, fish growth performance (final body weight (FBW), percent weight gain (PWG) and specific growth rate (SGR)) showed a quadratic effect (P < 0.01) with MOS supplementation, compared with the control group, the FBW, SGR and PWG of the optimal group (400 mg/kg MOS) were increased 21.59%, 16.24% and 31.34%, respectively; After A.hydrophila challenged, the survival rate of fish in all group was 100%, the enteritis morbidity and red-skin morbidity showed a quadratic effect with MOS supplementation (P < 0.05), compared with the control group, the enteritis morbidity and red-skin morbidity of the optimal group (400 mg/kg MOS) were decreased 53.45% and 42.56%, respectively [30,33].

We hope our answer can get your agreement and recognition. If you have any questions, please do not hesitate to contact me. Thank you again for your comments!

Point 2: Results:

Furthermore, in figure 2 and figure 3, this reviewer has the impression that 400 mg/kg MOS is more effective than 1000 mg/kg. Please discuss this point.

Response 2: Thank you very much for your kind comment. This is a very interesting consideration. Many studies also have found similar results. For this interesting phenomenon, we also through frequent in-depth discussion. We will discuss this issue in detail and present our speculations in the Discussion section by summarizing the data and relevant information we have reviewed.

Lines 461 to 483:

We speculate that the reason for this difference may be partly related to intestinal digestive function which is closely related to gut motility and chyme viscosity [7,60]. MOS is a kind of FODMAPs (fermentable oligosaccharides, disaccharides, monosaccharides, and polyols) [7]. Many studies have shown that adding high doses of FODMAPs to the diet can exacerbate diarrhea while adding low doses can significantly reduce diarrhea symptoms [7,61,62]. However, intestinal diarrhea is closely related to gut motility. Another study on rat gut showed that SCFAs could enhance gut motility, reduce intestinal chyme pass time and promote the process of evacuation [5,63]. Our previous study in grass carp intestine also confirmed that MOS supplementation increased the concentration of short-chain fatty acids (SCFAs) [30]. It has been reported in European sea bass intestine revealed that MOS supplementation could increase the number of goblet cells [23], which could secrete mucus to protect the intestinal epithelium, lubricate the chyme and guarantee the process of evacuation proceeds smoothly [64]. On the other hand, MOS also is a water-soluble fiber, which could swell the gastrointestinal tract and increases the viscosity of the digesta in mice [7]. This leads to an increase in satiety and a decrease in the intake absorption of nutrients in the gut. Our early growth performance data also showed that the 1000 mg/kg MOS group would lead to a decline in comparison with the 400 mg/kg MOS group [30]. Another study on zebrafish also found that excess MOS can decrease growth performance [65]. In summary of the above evidences, we believe that the feed of the high-dose MOS group was incomplete digested and absorbed, and the MOS was excreted from the gut directly without effective utilization. Therefore, the protective intestinal tight junctions’ effect of 400 mg/kg MOS is better than 1000 mg/kg MOS under A. hydrophila challenge conditions.

We hope our answer can get your agreement and recognition. If you have any questions, please do not hesitate to contact me. Thank you again for your comments!

Added References:

Hanau, S.; Almugadam, S.H.; Sapienza, E.; Cacciari, B.; Manfrinato, M.C.; Trentini, A.; Kennedy, J.F. Schematic overview of oligosaccharides, with survey on their major physiological effects and a focus on milk ones. Carbohydrate Polymer Technologies and Applications 2020, 1, doi:10.1016/j.carpta.2020.100013.

Vicentini, F.A.; Keenan, C.M.; Wallace, L.E.; Woods, C.; Cavin, J.B.; Flockton, A.R.; Macklin, W.B.; Belkind-Gerson, J.; Hirota, S.A.; Sharkey, K.A. Intestinal microbiota shapes gut physiology and regulates enteric neurons and glia. Microbiome 2021, 9, 210, doi:10.1186/s40168-021-01165-z.

Halmos, E.P.; Muir, J.G.; Barrett, J.S.; Deng, M.; Shepherd, S.J.; Gibson, P.R. Diarrhoea during enteral nutrition is predicted by the poorly absorbed short-chain carbohydrate (FODMAP) content of the formula. Aliment Pharmacol Ther 2010, 32, 925-933, doi:10.1111/j.1365-2036.2010.04416.x.

Halmos, E.P. Role of FODMAP content in enteral nutrition-associated diarrhea. J Gastroenterol Hepatol 2013, 28 Suppl 4, 25-28, doi:10.1111/jgh.12272.

Fukumoto, S.; Tatewaki, M.; Yamada, T.; Fujimiya, M.; Mantyh, C.; Voss, M.; Eubanks, S.; Harris, M.; Pappas, T.N.; Takahashi, T. Short-chain fatty acids stimulate colonic transit via intraluminal 5-HT release in rats. American Journal of Physiology-Regulatory, Integrative and Comparative Physiology 2003, 284, R1269-R1276, doi:10.1152/ajpregu.00442.2002.

Cheng, J.; Laitila, A.; Ouwehand, A.C. Bifidobacterium animalis subsp. lactis HN019 Effects on Gut Health: A Review. Front Nutr 2021, 8, 790561, doi:10.3389/fnut.2021.790561.

Paone, P.; Cani, P.D. Mucus barrier, mucins and gut microbiota: the expected slimy partners? Gut 2020, 69, 2232-2243, doi:10.1136/gutjnl-2020-322260.

Forsatkar, M.N.; Nematollahi, M.A.; Rafiee, G.; Farahmand, H.; Martínez-Rodríguez, G. Effects of prebiotic mannan oligosaccharide on the growth, survival, and anxiety-like behaviors of zebrafish (Danio rerio). Journal of Applied Aquaculture 2017, 29, 183-196, doi:10.1080/10454438.2017.1306732.

Point 3: Discussion:

How about the impact of MOS on other animal species?

Response 3: Thank you very much for your kind comment and question. Your question is very good. Actually, most studies on MOS on other species focus on the immune system, but the structural integrity of the study is scarce. We summarize the studies we collected as follows. Accodrding your helpful suggestion, we have added “4.4 The impact of MOS on other species' intestinal structural integrity” in Discussion section

Lines 559 to 578:

4.4 The impact of MOS on other species' intestinal structural integrity

Till now, there exist many studies on prebiotics and their impact on the intestinal immune system of terrestrial and aquatic animals, whereas few studies on intestinal structural integrity. As mentioned above, only two studies reported MOS can improve the AJCs (like Claudin-3 and E-cadherin) in fish (European sea bass and Rainbow Trout) [23,24]. In terrestrial animals, studies on broilers showed that MOS could reduce the concentration of serum DAO and endotoxin, and maintain intestine Occludin and Claudin-3 expression under enteritis model conditions [66,83]. Studies on mice ileum and Caco-2 cell displayed that MOS could up-regulate the expression of Claudin-1, ZO-1 and MUC-2 under LPS-induced conditions [67]. It has been reported that MOS could up-regulate ZO-1, Occludin and Claudin-1 (not jejunum) gene expression in jejunum and duodenum of weaned piglets [84]. In addition, there is a report that exhibited that MOS could enhance tight junctions by promoting transepithelial electrical resistance (TEER) in T-84 cells [85]. From the above literature, we can find that MOS has an excellent beneficial effect on the intestinal structural integrity of most species. However, studies on the intestinal structural integrity of MOS in humans, terrestrial and aquatic animals are not comprehensive enough. The current study not only verified the phenotypic destruction of intestinal structure in animals under pathogenic bacteria challenged conditions, but also systematically tight junction and adherent junction, and further revealed the potential pathways involved, providing a valuable reference for subsequent studies.

We hope our answer can get your agreement and recognition. If you have any questions, please do not hesitate to contact me. Thank you again for your comments!

Added reference:

Zhang, H.; Zhou, Y.; Xu, H.; Liang, C.; Zhai, Z. Bacillus amyloliquefaciens BLCC1-0238 Alone or in Combination with Mannan-Oligosaccharides Alleviates Subclinical Necrotic Enteritis in Broilers. Probiotics and Antimicrobial Proteins 2022, 14, 158-168, doi:10.1007/s12602-021-09853-w.

Song, M.; Fan, Y.; Su, H.; Ye, J.; Liu, F.; Zhu, X.; Wang, L.; Gao, P.; Shu, G.; Wang, Z.; et al. Effects of Actigen, a second-generation mannan rich fraction, in antibiotics-free diets on growth performance, intestinal barrier functions and inflammation in weaned piglets. Livestock Science 2019, 229, 4-12, doi:10.1016/j.livsci.2019.09.006.

Nopvichai, C.; Pongkorpsakol, P.; Wongkrasant, P.; Wangpaiboon, K.; Charoenwongpaiboon, T.; Ito, K.; Muanprasat, C.; Pichyangkura, R. Galactomannan Pentasaccharide Produced from Copra Meal Enhances Tight Junction Integration of Epithelial Tissue through Activation of AMPK. Biomedicines 2019, 7, doi:10.3390/biomedicines7040081.
